# Health-Related Quality of Life, Stress, Caregiving Burden and Special Needs of Parents Caring for a Short-Statured Child—Review and Recommendations for Future Research

**DOI:** 10.3390/ijerph20166558

**Published:** 2023-08-11

**Authors:** Lea Lackner, Julia Quitmann, Kaja Kristensen, Stefanie Witt

**Affiliations:** Department of Medical Psychology, Center for Psychosocial Medicine, University Medical Center Hamburg-Eppendorf, Martinistraße 52, W26, 20246 Hamburg, Germany

**Keywords:** health-related quality of life, stress, caregiving burden, special needs, parents, short-statured children

## Abstract

Children with short stature can experience a range of burdens due to their chronic condition. However, little is known about parents’ experiences dealing with their child’s short stature and the potential caregiving burdens and concerns they may face. We aim to review the literature on health-related quality of life (HRQOL), caregiving burden, and special needs among parents caring for a child with isolated growth hormone deficiency (IGHD) or idiopathic short stature (ISS). Using pre-defined inclusion and exclusion criteria, we systematically searched for literature using PubMed and Web of Science from its inception to December 2022. We identified 15 articles assessing HRQOL, special needs, or caregiving burdens in parents of IGHD/ISS children. The main problems included concerns about the future, organizational issues, side effects from growth hormone treatment, and social stigmatization. Furthermore, two studies assessed parents’ special needs to cope with caregiving stress, mainly the dialogue between them and their families or parent support groups. This review outlines parental burdens, needs, and resources when caring for an IGHD/ISS child. Furthermore, it provides information about previously used measures appraising parents’ special needs and underlines the need for disease-specific measurements.

## 1. Introduction

Short stature is defined as a disorder in which the height of an individual is more than two standard deviations (SD) below the corresponding mean height for a given age, sex, and population group. It is associated with many different diseases, such as genetic or endocrine ones [1], whereas the most common causes are growth hormone deficiency, hypothyroidism, Turner syndrome, and celiac disease [2]. Isolated growth hormone deficiency (IGHD) is an endocrine disease caused by a lack or insufficiency of growth hormone (GH) secretion [3] and therefore leads to short stature. Besides IGHD, many children fit the definition of short stature but have no underlying pathogenesis or etiology for being short [1,4]. In these cases, idiopathic short stature (ISS) is diagnosed [1]. In 2019, it was estimated that 144 million children worldwide under five years were short-statured, according to United Nations Children’s Fund (UNICEF) et al. [5]. A study that examined the data of the Pfizer International Growth Study (KIGS^®®^) from Europe, Asia, and Japan revealed that 46.9% of the sample with short stature (n = 83,803) exhibited short stature as a result of IGHD, while 8.2% presented with ISS [6]. This finding underscores the significance of IGHD and ISS in children within the medical system. Another study highlights that short stature is one of the most frequent concerns pediatric endocrinologists and other physicians caring for children must deal with [4]. Given that this topic affects many children and their families, several studies have been conducted to investigate the IGHD/ISS children’s well-being and health-related quality of life (HRQOL). This was also reinforced by the fact that the impact of chronic conditions on children and their families has gained importance in recent years [7,8]. Most studies assume that short-statured children have a lower HRQOL compared to their normal-statured peers [9] and have low self-esteem [10], primarily because of short-statured children being bullied at school. Worries about their height, feeling inferior about their shortness, and negative comparison with peers were additional results retrieved in the studies on IGHD/ISS children’s HRQOL [11,12,13].

A chronic pediatric health condition affects the child and the entire family [14,15]. Especially the parents of children with a chronic condition are often responsible for maintaining the functioning of the family by emphasizing the positive aspects of the development of their children and helping them cope with their chronic illness [16,17]. Socioecological factors such as a good functioning parent–child relationship and parental adaptation are some of the main protective factors for these children [18,19]. Moreover, studies have shown that high caregiving stress levels can affect children by causing depression, anxiety, and feeling desolate [20]. Therefore, it is crucial to investigate parental HRQOL, burdens, and special needs throughout their children’s short stature and to understand how they are affected by caring for a short-statured child because parents can modulate their children’s intrapersonal emotional attitude towards themselves with social support and coping strategies [21,22]. The better the parent can deal with the caregiving burden, the better the child will be able to cope with their health-related burdens [23]. Parents’ HRQOL is essential as it significantly affects the parent reports on their children’s HRQOL [21]. That is essential to keep in mind since parents’ perception of the well-being and functioning of their child might affect treatment decision making and healthcare utilization [24].

Although numerous studies have been conducted on short stature in general and its impact on children, only a limited number of studies have addressed the impact of pediatric IGHD/ISS on parents, and the results are inconclusive [25]. It is important to differentiate between the different etiologies of short stature, as they require different medical treatments and also have different prognoses. IGHD and ISS can be treated with GH injections, although this treatment option is only approved in the USA and may only be used off-label in Germany and other countries [26].

This research aims to review the current literature and find out how parents are challenged throughout caregiving for a child with IGHD/ISS. We also intended to find out about the parents’ HRQOL and what kind of special needs they might have. Another aim was to detect how the included studies assessed parents’ caregiving burden and HRQOL and if they used a disease-specific or generic approach. In this context, special needs mean the requirements parents might need to cope with the caregiving burden. HRQOL is defined as the subjective perception of health, including physical, social, and emotional well-being [27]. Over the years, it has become an outcome indicator in the medical field [28]. Furthermore, the caregiving burden can be defined as the strain on a person caring for a chronically ill or disabled family member [29]. Understanding the aspects of the caregiving burden is crucial as it is connected to the well-being of the individual and the caregiver [30]. Therefore, we aim to understand the aspects of the caregiving burden in short-stature youth and identify patient-reported outcomes measures (PROMs) assessing the caregiving burden and parental quality of life for use in routine care and research by conducting a scoping review.

## 2. Materials and Methods

We conducted a literature review in December 2022 using PubMed and Web of Science (Core Collection) without any limitation on the year of publication, the language employed, or the accessibility of full-text articles. Furthermore, sources in the included articles were searched for additional materials (hand searching). We followed the methodological framework of Arksey and O’Malley [31] for scoping reviews. This framework includes five stages, as well as an optional sixth stage: (1) identifying the research question; (2) identifying relevant studies; (3) study selection; (4) charting the data; (5) collating, summarizing, and reporting the results; and (6) a consultation exercise [31].

Publications up to the search date, including information about HRQOL, caregiving burden, or special needs of parents of IGHD/ISS children, were identified. Text word searches and Mesh-Terms, only available on PubMed, were used to avoid missing relevant articles. We used a combination of keywords and database-specific search terms. The search term included a combination of keywords and MeSH terms combined with the Boolean operators “AND” and “OR”: For the diagnosis ISS, we used the following term: “Growth Disorders”[Mesh] OR “Growth disorder*”[tw] OR “short stature”[tw] OR “idiopathic short stature”[tiab] OR “Dwarfism”[Mesh] OR “rare condition*”[tiab]; for the diagnosis of IGHD we used this term: “Dwarfism, Pituitary”[Mesh] OR “Dwarfism, Pituitary/psychology”[Mesh] OR “Hypopituitarism”[Mesh] OR “Pituitary insufficiency”[tiab] OR “Insulin-Like Growth Factor II/deficiency”[Mesh] OR “Insulin-Like Growth Factor I/deficiency”[Mesh] OR “Insulin-like growth factor-I Deficiency”[tiab] OR “Human Growth Hormone/deficiency”[Mesh] OR “Growth Hormone/deficiency”[Mesh] OR “Growth Hormone-Releasing Hormone/deficiency”[Mesh] OR “growth hormone deficiency” [tw]. Those terms were connected to the following terms with “AND”: “Parents”[Mesh] OR “Parents/psychology”[Mesh] OR “Caregivers”[Mesh] OR Parent*[tw] OR mother*[tw] OR father*[tw] OR caregiver*[tw] AND “Cost of Illness”[Mesh] OR “costs of illness”[tw] OR “Quality of Life”[Mesh] OR “quality of life”[tw] OR “parental quality of life”[tw] OR “Mental Health”[Mesh] OR “mental health”[tw] OR “Parent–child Relations”[Mesh] OR “Family Conflict”[Mesh] OR “well being “[tw] OR “well being “[tw] OR “emotional drain”[tw] OR “caregiving stress”[tw] OR “caregiving burden”[tiab] OR “parental burden”[tw] OR “parent reported outcome*”[tw] OR “psychosocial outcome”[tw] OR “psychosocial need*” [tw] OR “burden of disease”[tw] OR “health related quality of life”[tw] OR “health outcome”[tw].

The process of publication selection followed the PRISMA statement [32]. We used pre-defined inclusion and exclusion criteria to screen titles and abstracts (Figure 1). Included in this study were research papers that encompassed parents of children diagnosed with IGHD or ISS within the age range of 0–21 years. Studies examining parents’ QoL, mental health, or general well-being were considered for inclusion, as well as those investigating the parental burdens associated with their child’s chronic condition. Additionally, studies exploring parental needs and resources were included. Furthermore, the inclusion criteria involved peer-reviewed journals, cross-sectional studies, clinical trials, prospective studies, longitudinal studies, qualitative studies, and case reports. Excluded from this study were research papers that focused on parents of children with causes of short stature other than IGHD/ISS, such as achondroplasia, small for gestational age, Turner syndrome, skeletal dysplasia, or psychosocial dwarfism. The only exception was made for studies that included the largest number of participants who were parents of children with IGHD/ISS and evaluated them separately. Studies that solely compared treatment and non-treatment groups, focusing on the effectiveness of growth hormone therapy, were also excluded. Additionally, studies that solely examined the child’s HRQOL were excluded. Conference abstracts, reviews, and meta-analyses were also excluded from the study.

We removed duplicated publications after the initial search in December 2022 using the above-mentioned search term. In the next step, a title and abstract screening were conducted by one researcher. Two independent raters screened eligible full texts to ensure an unbiased selection. Outcomes sought for this review were caregiving burden, HRQOL, stress, and special needs of parents caring for an IGHD/ISS child.

We charted the data using Microsoft Excel (version 2016, Microsoft Corporation, Redmont, WA, USA) and identified the relevant data on the sample (population and country of origin), the study design used, the aim of the studies and the PROMs used, and the main results from all included publications.

## 3. Results

### 3.1. Study Characteristics

We included 15 publications (published 1979–2022) for qualitative synthesis [16,19,25,33,34,35,36,37,38,39,40,41,42,43,44] (Table 1).

Three studies resulting in four publications were conducted within the QoLISSY project [19,24,34,42], and another study with three publications was carried out by the Brod Group [35,40,43].

The publications were conducted in Germany [19,33,42], Italy [34,39], the United States of America (USA) [16,36], Spain [44], Poland [37], Sri Lanka [38], and the Netherlands [41]. Four publications were conducted multi-nationally [25,35,41,43].

The age of the children with ISS and IGHD included in the publications ranged from 4 to 18 years. Sample sizes ranged from n = 11 parents [16] to n = 243 parents [43]. Out of the 15 selected publications, 12 were cross-sectional, one utilized a prospective observational design, another used a non-interventional design, and the remaining used an uncontrolled before-and-after design. Seven of these publications were multi-center trials, and five were single-center trials. Eight publications approached parental burdens using a quantitative approach [25,33,36,37,38,42,43,44]. Moreover, one publication reported structured interviews to capture parental burdens [16]. Another publication utilized a mixed-method approach using questionnaires and interviews [41]. Two publications carried out focus group discussions [19,34]. Another two publications combined structured interviews and focus group discussions [35,40]. A narrative-based approach was implemented in one publication [39].

### 3.2. Parents’ HRQOL, Stress, Caregiving Burdens, and Special Needs

Casana-Granell, Lacomba-Trejo, Montoya-Castilla and Perez-Marin [44] examined stress levels and emotional distress at 145 principal caregivers of short-statured children, most likely ISS children aged 12–17.

This study utilized the Pediatric Inventory for Parents (PIP) to assess the caregivers’ stress levels, a chronic–generic questionnaire for parents with chronically ill children. In addition, this study used the Hospital Anxiety and Depression Scale (HADS) to evaluate the principal caregivers’ emotional distress by analyzing possible symptoms of anxiety and depression. According to the PIP, 44.2% to 62.7% of the principal caregivers showed high stress levels. The resistance and always-returning care situation especially contributed to these high stress levels (57.3% of centiles > 50).

Furthermore, 15.2% (n = 22) of the principal caregivers expressed a clinically significant problem with overall emotional distress on the HADS. Additionally, 47.6% (n = 69) showed anxiety symptoms (22.8% most likely had an anxiety disorder, 24.8% had a clinical anxiety disorder), and 17.2% seemed depressed. Casana-Granell, Lacomba-Trejo, Montoya-Castilla and Perez-Marin [44] emphasized that almost every principal caregiver was the mother of the short-statured child, which confirmed earlier findings [45]. The authors did not address the risk of bias within their study.

Majewska, Stanisławska-Kubiak, Wiecheć, Naskręcka, Kędzia and Mojs [37] assessed anxiety levels in 101 mothers of children with growth failure due to IGHD or unknown causes. The children were aged 5 to 16 years, and of them, 70 were diagnosed with IGHD and received human growth hormone (hGH), and 31 were undergoing the diagnostic process to determine the short stature etiology.

The Spielberger State–Trait Anxiety Inventory (STAI) was used to assess mothers’ anxiety levels. Anxiety as a trait was low in all recruited mothers; nevertheless, it was higher in mothers whose children did not get diagnosed yet. Anxiety as a state was presented with medium anxiety levels, whereas mothers without a diagnosed child showed higher values. Overall, the mothers of children without diagnosis or treatment presented significantly higher anxiety levels (*p* = 0.001). The risk of bias was not addressed by the authors. The small sample size and the limitation of only mothers of short children being included must be considered as limitations within this study.

The Brod Group recruited thirty-one parents of IGHD children aged between 4 and 13 years to develop a model of the impact of IGHD [35] and assess the burden of GHT on children and parents [35]. They conducted four focus group discussions and 52 telephone interviews with IGHD children and their parents/guardians.

Within this one study, the Brod Group published the three following papers.

Brod, Alolga, Beck, Wilkinson, Højbjerre and Rasmussen [35] aimed to develop a model of the impact of IGHD to support a disease-specific patient-reported outcome (PROM) and an observer-reported outcome (ObsROM) measure. Parents (n = 31) reported their emotional impacts from their children’s IGHD. Nearly half of the parents reported worry for their child (n = 16, 47%), anger or frustration over the reactions of others about their child’s size (n = 13, 38%), relief when a diagnosis was made (n = 10, 29%), and pressure in managing treatment for their children (n = 4, 12%). The limitations addressed by the authors were the generalizability of the findings. Although it was a large sample size for qualitative research, these findings may not be generalizable to parents of IGHD children in other countries or ethnic groups.

Brod, Højbjerre, Alolga, Beck, Wilkinson and Rasmussen [40] assessed the burden of GHT from the child’s perspective and the impact of the children’s treatment on the parents. Parents (n = 31) reported being emotionally impacted by hGH treatment for their children. Of these, 62% (n = 21) noted their worry, including worry about treatment administration (n = 20, 59%), causing pain to their child (n = 13, 38%), and medication costs (n = 5, 15%). The second treatment burden identified for parents was the “interference” domain. Half of the parents (n = 17, 50%) reported that hGH treatment interferes with family travels. Thirty-two percent (n = 11) of the parents noted that preparing their child for the injection took time. For 12% (n = 4) of parents, the hGH treatment interfered with their daily and social life. An attempt to minimize recall bias for those taking treatment was made by having a relatively short duration (no more than 12 months).

Brod, Rasmussen, Alolga, Beck, Bushnell, Lee and Maniatis [43] aimed to describe the psychometric validation data for three measures generated through a concept elicitation phase beforehand. The elicitation consisted of a literature review, interviews with clinical experts, four focus groups organized in Germany, and 52 telephone interviews in the UK and the US. The Growth Hormone Deficiency-Parent Treatment Burden Measure (GHD-PTB) was one of the three generated measures. The assessment tool is a PRO that evaluates the treatment burden experienced by parents/guardians of children aged 4 to <13 years with IGHD.

To conduct the psychometric validation for the GHD-PTB, n = 243 parents/guardians of IGHD children aged 4 to <13 years completed the GHD-PTB. Parent/guardian mean age was 41.6 years (range 22–66), with most coming from the US (91.8%). Of the parents/guardians, 80.7% were mothers, and 88.1% were married.

The average scores of the eight items in the GHD-PTB ranged from 0.35 to 1.28 on a response scale ranging from 0 (“Not at all/Never”) to 4 (“Extremely/All of the time”). In the group that started hGH treatment within this study, significant improvements were observed in the Emotional and Overall domains, with scores decreasing by 16.6 and 8.6 points, respectively, on a 0-to-100-point scale. One limitation addressed by the authors was the sample. Most participants were from the US and white, so the results may not be generalizable.

Marini, Chesi, Mazzanti, Guazzarotti, Toni, Salerno, Officioso, Parpagnoli, Angeletti, Faienza, Iezzi, Aversa and Sacchetti [39] aimed to understand illnesses of children and teenagers with IGHD and their families’ experiences through a narrative-based approach.

Some parents (n = 48) wrote about waiting for the diagnosis. Over half of them spent time living with anxiety and concerns (n = 26, 55%). Many mentioned the communication of hGH treatment in their narratives (n = 51 answers). One-third were worried and unconvinced about the treatment (n = 16, 30%). Most parents (n = 89 answers) wrote about the treatment difficulties: 33% (n = 29) had organizational issues because of the daily injections. For 21% (n = 19) of parents, it was painful to cause pain to their affected child. Asking the parents about their worries, over half of them (67%, n = 28) were worried about the side effects of hGH treatment, and 14% (n = 6) of parents feared the therapy would not work. No differences showed up throughout the parents’ narratives depending on the children’s gender. The authors did not address limitations within their study.

The QoLISSY project was a multi-center study conducted simultaneously in five European countries (France, Germany, Spain, Sweden, and the United Kingdom (UK)). They aimed to develop a disease-specific HRQOL instrument for children and adolescents aged 8–18 with IGHD/ISS and for parents of IGHD/ISS children aged 4–18. In total, three studies were conducted, which resulted in the following four publications.

Quitmann, Rohenkohl, Sommer, Petzold and Bullinger-Naber [19] executed focus-group discussions with item generation for the QoLISSY project in Germany.

The study aimed to identify important aspects of children’s HRQOL from parents’ and children’s perspectives. Although the primary aim was to focus on children’s HRQOL, parents expressed personal burden due to their child’s short stature. Specifically, parents described everyday problems concerning their child’s height and future anxieties concerning the short body height of their children. Problems in treatment were another burden for parents of short-statured children, and the treatment organization appeared to be a main concern. Social stigmatization also played a significant role in talking about hGH treatment. Parents were frustrated by the lack of knowledge of others.

Silva, Bullinger, Sommer, Rohenkohl, Witt and Quitmann [25] aimed to compare the levels of caregiving stress and HRQOL of short-statured children’s parents between different clinical groups of diagnosis, treatment status, and current height deviation. Furthermore, the authors examined the direct and indirect links between children’s psychosocial functioning and their parents’ HRQOL by using caregiving stress as an indicator.

The authors took advantage of the “effects on parents” scale of the QoLISSY questionnaire to assess caregiving stress. In total, parents of children/adolescents with ISS reported greater caregiving stress than parents of IGHD children. Parents of children with current short stature reported greater caregiving stress than those who achieved normal height.

A significant effect of children’s psychosocial functioning was found on caregiving stress (ß = −0.53, *p* < 0.01). According to this, caregiving stress directly affected parents’ QoL (ß = −0.37, *p* < 0.01). The indirect effect of children’s psychosocial functioning on parents’ QoL via caregiving stress was statistically significant (ß = 0.20, *p* < 0.01; BC 95% CI = 0.11/0.31).

Bloemeke, Silva, Bullinger, Witt, Dörr and Quitmann [42] conducted a prospective observational study to evaluate the QoLISSY questionnaire as a health-outcome indicator of human growth hormone (hGH) interventions.

The authors used the generic KIDSCREEN-10 and the QoLISSY questionnaire. Raw QoLISSY scores were transformed into 0 to 100 scores, whereas higher values represented a higher HRQOL.

Comparing the treated and untreated group at baseline on the QoLISSY questionnaire showed that parents of untreated children reported a significantly higher HRQOL in the domain “future” than parents of children in the treated group (parents of treated children (IGHD/SGA): M = 53.93, parents of untreated children (ISS): M = 68.57).

Overall, the results displayed a trend of HRQOL scores improving in all QoLISSY domains in the treated sample while decreasing in the untreated sample. These differences were not statistically significant. Furthermore, the generic KIDSCREEN-10 instrument could not detect changes in HRQOL, whereas the QoLISSY questionnaire detected changes in HRQOL between the treated and untreated groups in this sample. Endocrine short stature is a rare disease; the sample size was large but still very selective because they were all looking for treatment options in the growth clinics. So the sample might not be representative of the overall target population.

Quitmann, Giammarco, Maghnie, Napoli, Di Giovanni, Carducci, Mohn, Bullinger and Sommer [34] adapted and validated the existing QoLISSY questionnaire for Italian patients and parents by undergoing focus group discussion and a cognitive debriefing process.

“Effects on parents” was the second-largest category, with about 20% of codes. Parents described the effect of their children’s growth deficit as being sad and wishing for a better life for their children. Fifteen percent of the parents addressed hGH treatment as a burden of treatment administration or expressed concerns about possible side effects. Limitations within this study were the small sample size due to unforeseen difficulties in recruitment, which makes generalizing the results to other ethnic groups difficult.

Visser-van Balen, Geenen, Kamp, Huisman, Wit and Sinnema [41] assessed parental stress about their child’s future and their worries about the children’s psychosocial functioning to understand parents’ motives for choosing hGH treatment. A psychologist interviewed parents about their consideration of their child having equal chances in the labor market compared to people of normal height and the children’s prospect of finding a spouse (yes, doubtful, no), and 44.5% of the parents expected their child to have a lower prospect in the labor market as an adult (39% of boys, 48% of girls), while 39% of the parents considered their child to have a lower prospect of finding a spouse (77% of boys, 17% of girls). This difference between boys and girls was significant (*p* < 0.01).

Furthermore, the CBCL was utilized in this study. Two out of three parents reported worries about future opportunities or observed psychosocial problems in their children. One limitation of this study is the sample size. It was sufficiently large to conclude with a comparison with normative data but too small to examine other possible roles in modulating the results with sufficient power.

Hitt, Ginsburg, Cousounis, Lipman, Cucchiara, Stallings and Grimberg [36] explored factors influencing parental height-related decision making to seek medical treatment for their child. Using a five-point Likert scale, parents answered the following question: “how much of an impact would each of the following issues make on your decision whether to do something medical for your child’s height?”.

Most parents were concerned about the efficacy and side effects of hGH treatment. This was illustrated by 64% of the parents rating “treatment characteristics” as having a big or extreme impact on their decision to seek medical treatment for their child’s short stature. Sixty percent of the parents were concerned that their child’s height was short relative to their peers and demonstrated a strong focus on external comparisons of their children to others. In third place came the category “health”, rated by 54% of the parents as having a significant impact on their decision. The survey consisted of two additional questions to assess the parents’ opinions on how much hGH treatment could improve their HRQOL. Seventy-six percent of the parents rated hGH treatment as potentially improving any QoL issue related to their child’s short height. The authors mentioned a response bias (the desire to give the socially preferred answer) within their study, which may have contributed to a reduced rating of specific categories of concern.

De Silva and de Zoysa [38] assessed mental health difficulties in 74 parents of children with IGHD. The authors used the General Health Questionnaire-30 (GHQ-30) to examine the parents’ mental health. A score of 4 or above identifies the respondent as having mental health difficulties.

Fifty-four percent (n = 40) of the parents with an IGHD child scored at or above 4, indicating that more than half the parents showed evidence of mental health difficulties. Seventy percent (n = 28) of them were mothers. A limitation of this study was the small sample size and hence the generalizability and the statistical power of the study are limited. Furthermore, the results were preliminary, so no conclusion on causality can be made. On top of that, the GHQ-30 has not been validated in Sri Lanka. Because de Silva and de Zoysa only focused on parents’ mental health as an outcome and reported very concisely on their results, the results section of this study is brief.

Haverkamp and Noeker [33] assessed the psychosocial stress factors of parents associated with short stature. The authors invented a new disease-specific parental questionnaire to inquire about the parents’ short-stature-associated stress factors. The questionnaire includes four scales: “suffering”, “future anxieties”, “behavioral problems”, and “coping efforts”.

In general, the parents scored low on all four scales. Of the parents of IGHD children, 21.5% were anxious about possible hGH treatment side effects. Parents reported being stressed by future anxieties concerning their child’s professional career (13.9%). On top of that, the primary sources of energy for parents of IGHD children were good relationships with medical staff (38%) and medical intervention (35.4%). Another confidence booster named by 20.3% of the parents was their family. The authors controlled for parental misattributions using background information such as socioeconomic status.

Rotnem, Cohen, Hintz and Genel [16] investigated the subjective experiences and special needs of parents whose children failed to grow normally due to IGHD.

All parents expressed confusion about how to relate to their short-statured child. Most parents had been aware of their tendency toward overprotectiveness and not-age-appropriate expectations toward their children. However, they continued to react to their children according to size rather than age. It was more difficult for mothers and fathers to accept pathological short stature in boys than girls. Defenses most often used by these parents were denial, rationalization, and reaction formation through incorporation with Human Growth Foundation, Little People of America, or parent support groups. Limitations within this study include the small sample size of families, making generalization difficult.

### 3.3. Validated Tools to Assess Parents’ HRQOL, Stress, Caregiving Burdens, and Special Needs

In the studies identified and summarized above, eight different tools were used to assess parental HRQOL, caregiving burden, and special needs of parents with short-statured children; five of them were generic (PIP, HADS, STAI, GHQ-30, and EUROHIS-QOL-8) and three short-stature-specific (GHD-PTB, QoLISSY, and “Short stature in children—a questionnaire for parents”).

The PIP assesses 42 potentially stressful situations for parents of children with chronic illnesses. It measures the difficulty and frequency of each situation across four domains: medical care, communication, emotional functioning, and role function. Scores are obtained for each subscale by adding up the item scores. The PIP showed high internal consistency reliability using Cronbach’s alpha between 0.80 and 0.96. Its origin was developed in a sample of pediatric oncology patients in English in 2001 [46]. The instrument has been used in studies focusing on various illnesses, including, e.g., diabetes [47], cancer [48], short stature [20], and congenital malformations [49]. The PIP has been translated into, e.g., German [50] and Spanish [51].

The HADS, developed in 1983, is a tool used to assess anxiety and depression in a general medical population and consists of seven items for anxiety and seven items for depression [52]. Cronbach’s alpha for the subscale of anxiety (HADS-A) ranges from 0.68 to 0.93, and for the subscale of depression (HADS-D), Cronbach’s alpha varies between 0.67 and 0.90 [53]. Cut-off values are available for both scales. The HADS is available in many languages, e.g., German [54], Italian [55], Chinese [56], or Arabic [57].

The STAI is a psychological assessment comprising 40 self-report items on a four-point Likert scale. It measures two types of anxiety: state anxiety and trait anxiety. The inventory is divided into separate sections for each type, with 20 questions dedicated to each. Higher scores on the inventory indicate higher levels of anxiety. The STAI is available in over 40 languages [58], e.g., German [59], Spanish [60], and French [61]. Cut-off values are available [62]. The STAI demonstrates high internal consistency, with median alpha coefficients of 0.93 for state anxiety and 0.90 for trait anxiety [63].

The GHQ is a tool for assessing general mental health and well-being in non-psychiatric populations. The GHQ measures psychological distress through subscales such as somatic symptoms, anxiety and insomnia, social dysfunction, and severe depression. It comprehensively assesses an individual’s mental health status [64]. The reliability and validity of the GHQ-30 show values of Cronbach’s alpha of 0.95 for the GHQ-30. The GHQ is available in different versions consisting of 12, 28, 30, or 60 items and different languages, e.g., Spanish [65], German [66], and Italian [67].

The EUROHIS-QOL eight-item index, developed by the WHOQOL group, serves as an economic screening measure. It has been validated using data from multiple European countries, including France, Germany, the United Kingdom, Lithuania, Latvia, Croatia, Romania, Slovakia, the Czech Republic, and Israel. The index is conceptually derived from the original WHOQOL-BREF, with two items selected from each domain (physical, psychological, environmental, and social). The study’s findings revealed consistent and satisfactory internal consistencies across the countries studied [68,69].

The GHD-PTB is an eight-item PROM for parents/guardians of children aged 4 to <13 years with IGHD. The score is calculated by summing the individual item scores and converting them into a standardized score ranging from 0 to 100 points, where a higher score demonstrates a higher caregiver burden. The eight items result in two domains (parental emotional and parent interference) and one total score. Internal consistency reliability was acceptable and ranged between 0.60 and <0.70. The GHD-PTB is currently available in the English language [43].

The QoLISSY questionnaire is a PROM to assess the HrQol of short-statured children using self-reports (8–18 years) and proxy-reports (4–18 years) and consists of 50 items resulting in six subscales. The proxy version comprises two additional subscales: future (5 items) and effects on parents (11 items) [70]. Cronbach’s alpha varies between 0.65 and 0.95 [42]. The QoLISSY questionnaire is validated for ISS, IGHD [70], achondroplasia [71], and Small-for-Gestational Age [72] and is available in various languages, e.g., German, Englisch, Spain, Frech, Swedish [70], Italian [34], or Greek [73].

The questionnaire “Short stature: a questionnaire for parents” was developed by [33] in a sample of parents of patients with IGHD, achondroplasia, Turner syndrome, familial short stature, and constitutional delay of puberty and growth. This tool consists of 34 items, resulting in four scales (Suffering, Future anxieties, Behavioral problems, and Coping efforts). Cronbach’s alpha ranges from 0.60 to 0.91.

## 4. Discussion

Parenthood represents an incisive and momentous experience. The transition to parenthood will be assumed not only as a crisis but also as a normative event; it is often underestimated [74]. It has to be noticed that parenthood is a decades-long developmental task, resulting in various challenges. Parents/primary caregivers of young children especially experience a relinquishment of autonomy, personal liberty, occupational identity, and social and leisure activities [75]. In addition to increased requirements [51], mothers report multiple changes in their lives, which are experienced as stressful [76,77,78]. These challenges include mental health costs, such as time, physical and emotional well-being, conflicts of social roles, and economic restrictions [76,77,78]. Additionally, aspects of, e.g., the changes in responsibility and feelings of loss [79] were mentioned.

With the diagnosis of a child’s chronic health condition, parents/primary caregivers are confronted with additional responsibilities and tasks that may affect their quality of life [23,80]. Parents/primary caregivers of a child diagnosed with a chronic health condition may experience higher levels of emotional pressure than parents caring for a healthy child. Next to the primary responsibilities of parents/primary caregivers, they have to deal with additional disease-specific tasks to enable the healthy development of their child. Some studies examining the situation of families with a pediatric chronic disease highlighted that these parents have to adjust to restrictions in their social life, job perspective, and psychosocial well-being as their everyday life is decisively determined by the responsibilities for the child’s care [23,80]. The parental adaptation to the new situation of having a chronically ill child often happens without much support for coping with the child’s diagnosis. Supplies of psychosocial support are rarely offered, making it hard to assimilate the new family situation [80].

Children with IGHD/ISS do not need regular invasive medical interventions except daily injections, which is why their parents’ burdens are often questioned and minimized [44]. Therefore, this review investigated the caregiving burden, special needs, HRQOL, and stress in parents of IGHD/ISS children. The 15 included publications used various approaches to assess these domains; questionnaires, narrative-based approaches, structured interviews, and/or focus group discussions. Some authors also assessed the parents’ main sources of energy [16,19,33].

Most publications utilized questionnaires to assess the parents’ caregiving burden, special needs, HRQOL, and stress [25,33,36,37,38,42,43,44]. Various questionnaires were used, but most were generic and thus not sensitive enough to cover disease-specific aspects. While generic instruments are extensively validated and enable comparisons among different populations, they frequently fall short in identifying subtle yet clinically meaningful changes in HRQOL over time. This limitation arises from the absence of disease-specific aspects in the lives of affected patients that significantly influence their HRQOL [42]. Hitt, Ginsburg, Cousounis, Lipman, Cucchiara, Stallings and Grimberg [36], and Haverkamp and Noeker [33] engendered disease-specific questionnaires to assess parents’ concerns. Unfortunately, both only covered small domains, such as factors influencing parents’ decision-making for hGH treatment [36]. Bloemeke, Silva, Bullinger, Witt, Dörr and Quitmann [42] and Silva, Bullinger, Sommer, Rohenkohl, Witt and Quitmann [25] used the disease-specific QoLISSY questionnaire. Brod, Rasmussen, Alolga, Beck, Bushnell, Lee and Maniatis [43] implemented the disease-specific PRO GHD-PTB. Moreover, they developed two disease-specific measures for assessing children’s treatment burden. Structured interviews and focus group discussions were implemented in several publications [16,19,34,35,40,41]. Some developed a semi-structured interview guide [15,28,30,35] but still focused on different domains of parental concerns.

The QoLISSY Group and the Brod Group used the same interview guide for their studies, making the results more comparable.

The included publications reported various burdens, special needs, and HRQOL of the parents. On the one hand, hGH treatment and all the issues coming along with it were expressed as having a massive impact on the parents and their families. Organizational issues were especially mentioned by the parents/primary caregivers as a main burden [34,35,36,39,40]. Nevertheless, most parents still decided on hGH treatment for their children because the suggested benefits outweighed the possible side effects and daily injections. On the other hand, Brod, Rasmussen, Alolga, Beck, Bushnell, Lee and Maniatis [43] found that most parents had little to no problems with their children’s hGH treatment. Thus, the results on parental burden through hGH treatment are inconsistent and need further exploration.

Furthermore, parents mentioned future anxieties as being a major concern. Many were stressed about their children’s future because they believed that short-statured children had a lower prospect in the labor market and of finding a spouse as an adult due to their short stature [19,33,34,40].

Social stigmatization and comparison of height to their children’s peers appeared to be another caregiving burden for parents of IGHD/ISS children [19,35,36]. Some parents wished for psychological treatment besides the medical treatment option and were convinced they would have fewer problems if those in their environment knew about the parents’ and their children’s special needs and opportunities [19]. There were concerns that parents seemed to have high stress levels and symptoms of anxiety [37,44] and that the children’s psychosocial functioning significantly affected parents’ HRQOL via caregiving stress [24].

Many publications did not address parents’ special needs to cope with caregiving stress. Nevertheless, the dialogue between the parents and their families or parent support groups appeared to be a good coping mechanism for parents [12,20]. This domain needs to be explored in depth in further explorations.

Parents of children diagnosed with IGHD/ISS can have an impaired HRQOL, have special needs, and face various caregiving burdens and stress. The adaptation of these parents has a significant impact on other family members and their well-being, but parents need time and resources to adapt to a child’s chronic disease [81]. Once they have adapted, parents can increase their children’s well-being, and health-related outcomes may improve [23]. A crucial risk factor for parents’ impaired HRQOL is caregiving stress, which links the positive association between children’s psychosocial functioning and parents’ outcomes [25]. Besides stress mechanisms, a child’s development can be influenced by the feedback children with chronic diseases receive from their families [82,83]. Psychological support could be a huge benefit for the parents [32], but it is essential to discern those for whom this will be necessary to ensure positive feedback.

A further-developed disease-specific measurement besides the parent domain of the QoLISSY questionnaire [19] and the GHD-PTB [43] would be beneficial to find out more about the parents’ special needs, burdens, and HRQOL in the care of an IGHD/ISS child. If those dimensions are assessed and the parents are offered psychological counseling, their children could also benefit from the changed perception of their parents if needed. As earlier findings have shown, parental frustration can be transferred to their children [20,84]; in addition, good family functioning appears to be an essential protective factor for short-statured children [18].

### Limitations

The limitations of this study include the fact that we only used two databases for our search. While we believe that we identified all relevant articles through our search, we cannot definitively exclude the possibility that including additional databases would have yielded further results. Additionally, we did not assess the quality of the studies, making it challenging to compare their findings. Moreover, the included studies had very different designs that were not directly comparable. Furthermore, the included studies primarily focused on mothers, making it difficult to generalize the results to parents in general. Comparing the results is complicated because the studies were conducted in different healthcare systems. Cultural differences are also likely to influence the assessment of burdens and concerns. Most publications were from Europe and the USA, with only one from Sri Lanka. Another limitation of our review is that the included studies employed different recruitment strategies, often purposive sampling, which introduces a potential bias. Nonetheless, our review highlights parental burdens in dealing with IGHD/ISS children and draws attention to the limited research on this topic.

## 5. Conclusions

Parenthood itself is a decades-long task with various challenges. Parents/primary caregivers of children with chronic health conditions face additional challenges. While IGHD and ISS are not life-threatening health conditions, these challenges are often understated, and parents/primary caregivers rarely seek help in processing the diagnosis and in day-to-day tasks. In contrast, this review points out that parents of IGHD/ISS children can have various burdens, anxiety, special needs, and an impaired HRQOL in caring for their short-statured child. Clinicians should remember that parental support may positively affect the child’s development.

On top of that, our results support earlier findings concerning the limited use of disease-specific measures. Still, most measurements are generic and hence not very sensitive. However, the Brod Group has contributed towards disease-specific measures for parents of IGHD/ISS children with the development of the GHD-PTB. Further studies should aim to develop measures for all domains of affected parents’ lives. For children, especially those with a chronic condition, family support is a major protective factor and helps children cope with their disease. Therefore, assessing the caregiving burden, HRQOL, and special needs of parents and families of IGHD/ISS children is crucial, because once the parents/families are adapted to the child’s short stature, they can increase their child’s health-related outcomes and well-being.

## Figures and Tables

**Figure 1 ijerph-20-06558-f001:**
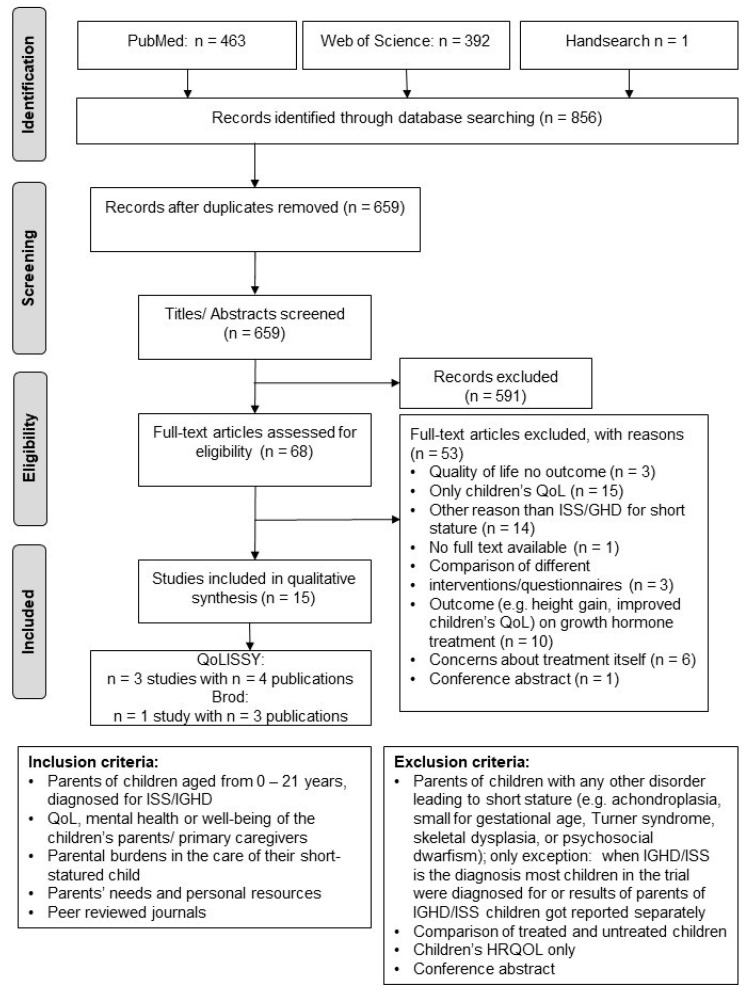
Prisma Flow Chart [32], search date 17 December 2022.

**Table 1 ijerph-20-06558-t001:** Overview of the included 15 publications.

Authors	Population; Country	Study Design	Aim of the Study and Measure Method	Results
Casana-Granell, Lacomba-Trejo, Montoya-Castilla and Perez-Marin [44]	145 principal caregivers of short-statured children aged 12–17 years; Spain	Cross-sectional study	Measuring stress levels of parents using PIP and their emotional distress using HADS	HADS: 47.6% showed symptoms of anxiety, 17.2% showed symptoms of depression. PIP: 44.2–62.7% showed high stress levels concerning caregiving.
Majewska, Stanisławska-Kubiak, Wiecheć, Naskręcka, Kędzia and Mojs [37]	101 mothers of children aged 5–16 years with IGHD or unknown cause for growth failure; Poland	Cross-sectional study	Assessing anxiety levels in mothers using the STAI	Trait anxiety: low in all recruited mothers; state anxiety: medium levels; mothers of children without diagnosis presented significantly higher levels of anxiety than mothers of children without diagnosis, as did mothers of younger children.
Brod, Rasmussen, Alolga, Beck, Bushnell, Lee and Maniatis [43]	243 parents of IGHD children aged 4 to <13 years; UK and the USA	Non-interventional	Describing the psychometric validation data for the three measures, namely GHD-CTB-Child, GHD-CTB-Observer, and GHD-PTB	Mean scores of the GHD-PTB ranged from 0.35 to 1.28. Treatment-naïve participants improved for the Emotional and Overall domains (−16.6 and −8.6 points) after 12 weeks.
Brod, Alolga, Beck, Wilkinson, Højbjerre and Rasmussen [35]	31 parents of children aged from 4–13 years with IGHD; Germany, the UK, and the USA	Cross-sectional study	Focus groups or telephone interviews guided by a semi-structured interview guide to understanding emotional well-being of parents of IGHD children	47% of parents worried for their child, 38% felt angry/frustrated over the reaction of others, 29% felt relieved receiving diagnosis, 12% were pressured in parenting and managing treatment for their child.
Brod, Højbjerre, Alolga, Beck, Wilkinson and Rasmussen [40]	31 parents of IGHD children aged 4–13 years treated with growth hormone; Germany, the UK, and the USA	Cross-sectional study	Focus groups or telephone interviews guided by a semi-structured interview guide to explore the parents’ burden of hGH treatment	Parents worried about hGH treatment administration (59%), causing pain to their child (38%), and medication costs (15%); 12% were affected in daily activities; 50% felt limited in family travel, 32% needed time to prepare child for injection.
Marini, Chesi, Mazzanti, Guazzarotti, Toni, Salerno, Officioso, Parpagnoli, Angeletti, Faienza, Iezzi, Aversa and Sacchetti [39]	72 parents of children aged 8–17 years with IGHD; Italy	Cross-sectional study	Narrative-based approach to collect stories from parents to understand their points of view	“Waiting for diagnosis”: 55% lived with anxiety; “difficulties of hGH treatment”: 33% had organizational issues, for 21% it was painful task; “expressed worries”: 67% worried about side effects of hGH treatment.
Quitmann, Rohenkohl, Sommer, Petzold and Bullinger-Naber [19]	31 parents of IGHD/ISS children aged 4–18 years; Germany	Cross-sectional study	Developing measurement tool to capture HRQOL in IGHD/ISS children and the view of the parents with focus group discussions	Main burdens expressed: comparison of height, everyday problems, future anxieties, and social stigmatization through others.
Silva, Bullinger, Sommer, Rohenkohl, Witt and Quitmann [25]	238 parents of IGHD/ISS children aged 8–18 years; France, Germany, Spain, Sweden, and the UK	Cross-sectional study	Levels of caregiving stress and HRQOL of parents were raised by parents reporting on their HRQOL via EUROHIS-QOL-8 Index and caregiving stress via QoLISSY scale “effects on parents”	Parents of currently short-statured children: greater caregiving stress than parents of children with normal height; significant indirect effect of children’s psychosocial functioning on parents’ HRQOL via caregiving stress (*p* < 0.01).
Bloemeke, Silva, Bullinger, Witt, Dörr and Quitmann [42]	Parents of children aged 4–18 years with IGHD (n = 65) or SGA (n = 58) starting hGH treatment; ISS children (n = 31) and the parents served as he control group (T0: n = 152 parents; T1: n = 126 parents); Germany	Prospective observational study	Evaluating the QoLISSY questionnaire as a health-outcome indicator of hGH interventions by assessing HRQOL before the start of hGH treatment (baseline, T0) and at 12 months after the start (T1) with KIDSCREEN-10 Index and QoLISSY questionnaire	QoLISSY questionnaire detected changes in parents’ HRQOL between treated and untreated patients. Parents of untreated children: higher HRQOL at T0 in domain “future” than parents of children in treated group; improvement on “effects on parents” from T0 to T1 for both intervention and control groups on QoLISSY.
Quitmann, Giammarco, Maghnie, Napoli, Di Giovanni, Carducci, Mohn, Bullinger and Sommer [34]	20 parents of IGHD/ISS children aged 4–18 years; Italy	Cross-sectional study	Testing the QoLISSY questionnaire in Italy by undergoing focus group discussions and a cognitive debriefing-process	“Effects on parents”: second-largest category with about 20% of codes. HGH treatment organization was main concern; 10 out of 31 parents expressed future anxieties.
Visser-van Balen, Geenen, Kamp, Huisman, Wit and Sinnema [41]	38 parents of children aged 11–13 years with ISS (n = 26) or SGA (n = 12); Netherlands	Cross-sectional study	Structured interviews and CBCL to understand the motives of parents for choosing hGH treatment	Two out of three parents: worries about future opportunities; 44.5% expected their child to have lower prospects in the labor market; 39% expected their child to have a lower prospect of finding a spouse.
Hitt, Ginsburg, Cousounis, Lipman, Cucchiara, Stallings and Grimberg [36]	166 parents of children with IGHD/ISS (6–16 years) seeking EP; USA	Cross-sectional study	Exploring factors that influence parental decision making to seek hGH treatment for their short-statured child by utilizing a survey	Efficacy and side effects concerned most parents (64%), 60% were concerned about comparison of their child with others, 54% about health in general.
de Silva and de Zoysa [38]	74 parents of children aged 8–18 years with IGHD; Sri Lanka	Cross-sectional study	Gathering mental health difficulties of parents using the GHQ-30	54% of parents seemed to have mental health difficulties, 70% of them were mothers.
Haverkamp and Noeker [33]	442 parents of children (mean age 10.6 years) with different pathological growth disorders (IGHD n = 79, ACH n = 47, TS n = 225, FSS n = 38, CDPG n = 53); Germany	Cross-sectional study	Testing new questionnaire for parents of short-statured children with four dimensions: “suffering”, “future anxieties”, “behavioral problems”, and “coping efforts”; comparison between IGHD and ACH	Parents of IGHD children scored low on all scales; 15.2% feared secondary psychological problems due to hGH treatment, future anxieties stressed 13.9%. Parents’ main sources of energy: good relationships with medical staff, medical intervention, and family.
Rotnem, Cohen, Hintz and Genel [16]	Families of hypopituitary IGHD children with a mean age of 11.3 years (n = 11 parents); USA	uncontrolled before-and-after study	Structured and open-ended interviews with parents before and during one year of hGH treatment about experiences and resources	Doubts about maternal competence and overprotectiveness. Defenses: denial, rationalization, and reaction formation through incorporation with organizations.

Abbreviations: IGHD = isolated growth hormone deficiency; ACH = achondroplasia; TS = Turner syndrome; FSS = familial short stature; CDPG = constitutional delay of puberty and growth; hGH = human growth hormone; ISS = idiopathic short stature; QoLISSY = Quality of Life in Short Stature Youth; PIP = Pediatric Inventory for Parents; HADS = Hospital Anxiety and Depression Scale; HRQOL = health-related quality of life; EP = endocrine subspecialist care; STAI = Spielberger State-Trait Anxiety Inventory; GHQ-30 = General Health Questionnaire-30; SGA = small for gestational age; CBCL = Child Behavior Checklist; UK = United Kingdom; USA = United States of America; GHD-CTB-Child = Growth Hormone Deficiency-Child Treatment Burden Measure; GHD-CTB-Observer = Growth Hormone Deficiency-Child Treatment Burden Measure-Observer: GHD-PTB = Growth Hormone Deficiency-Parent Treatment Burden Measure.

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
