# Peer review of "Health-Related Quality of Life, Stress, Caregiving Burden and Special Needs of Parents Caring for a Short-Statured Child—Review and Recommendations for Future Research"

_ijerph, 2023, doi:10.3390/ijerph20166558_

Round 1

Author Response

Reviewer 1:

Reviewer comment # 1: The manuscript gives clear picture in understanding the caregiving burden of the parents of children with short stature. The following comments are made to improve the quality of the study for publication.

Author's answer: Dear Reviewer, thank you for your feedback on our review. Following your suggestions, we have optimized the manuscript in the following manner.

Reviewer comment # 2: Introduction: Correct the sentence on line 27 and 28 to give clarity on linking Isolated growth hormone deficiency (IGHD) to short stature.

Author's answer: Dear Reviewer, thank you for your critical review. We have optimized the sentence to give more clarity. The respective section now reads (lines 33 to 37): " It is associated with many different diseases, like genetic or endocrine ones [1], whereas the most common causes are growth hormone deficiency, hypothyroidism, Turner syndrome, and celiac disease [2]. Isolated growth hormone deficiency (IGHD) is an endocrine disease caused by a lack or insufficiency of growth hormone (GH) secretion [3] and therefore leads to short stature. "

Reviewer comment # 3: Introduction: Correct the statement in line 36 to 40 to give better understanding.

Author's answer: Dear Reviewer, thank you for your feedback. We have corrected the sentence. It now reads (lines 48 to 56): "Given that this topic affects many children and their families, several studies were conducted to investigate the IGHD/ISS children's well-being and health-related quality of life (HRQOL). This was also reinforced by the fact that the impact of chronic conditions on children and their families has gained importance in recent years [7, 8]. Most studies assume that short-statured children have a lower HRQOL compared to their normal-statured peers [9] and have low self-esteem [10], primarily because of short-statured children being bullied at school. Worries about their height, feeling inferior about their shortness, and negative comparison with peers were additional results retrieved in the studies on IGHD/ISS children's HRQOL [11-13]. "

Reviewer comment # 4: Introduction: Rearrange the statement on line 46-47 to give better understanding.

Author's answer: Dear Reviewer, thank you for your hint on this section. We have rearranged the statement (lines 58 to 68): " Especially the parents of children with a chronic condition are often responsible for maintaining the functioning of the family by emphasizing the positive aspects of development of their children and helping them cope with their chronic illness [16, 17]. Socioecological factors like a good functioning parent-child relationship and parental adaptation are some of the main protective factors for these children [18, 19]. Moreover, studies have shown that high caregiving stress levels can affect children by causing depression, anxiety, and feeling desolate [20]. Therefore, it is crucial to investigate parental HRQOL, burdens, and special needs throughout their children's short stature and to understand how they are affected by caring for a short-statured child because parents can modulate their children's intrapersonal emotional attitude towards themselves with social support and coping strategies [21, 22]."

Reviewer comment # 5: Materials and Methods: Elaborate on statement 74 in relation to the statement in line 71 regarding without limitations.

Author's answer: Dear Reviewer, thank you for your critical review. We have corrected the statement as follows (see lines 96 to 100): "We conducted a literature review in December 2022 using PubMed and Web of Science (Core Collection) without limitation to the year of publication, the language employed, or accessibility of full-text articles. Furthermore, sources in the included articles were searched for additional material (hand search). We followed the methodological framework of Arksey and O'Malley [31]."

Reviewer comment # 6: Materials and Methods: Provide clarity in line 76 and consider removing the brackets. Elaborate without using the brackets.

Author's answer: Dear Reviewer, thank you for the hint on this line. We have removed the brackets (lines 102 to 104): " Text word searches and Mesh-Terms, only available on PubMed, were used to avoid missing relevant articles. We used a combination of keywords and database-specific search terms. "

Reviewer comment # 7: Materials and Methods: Clarify LL and SW bracketed words in lines 79 and 80.

Author's answer: Dear Reviewer, thank you for the indication. We have clarified these lines (see lines 145 to 146): "Two independent raters (Lea Lackner and Stefanie Witt) screened eligible full texts to ensure an unbiased selection."

Reviewer comment # 8: Results: The detail report of each study included is presented.

Author's answer: Dear Reviewer, thank you for this feedback. We intended to present the results precisely and clearly.

Reviewer comment # 9: Funding – please clarify the statement in line 534

Author's answer: Dear Reviewer, thank you for this hint. We have clarified the statement (see line 645): "This research received no external funding.”

Reviewer 2 Report

Health-related Quality of Life, Stress, Caregiving Burden and 2 Special Needs of Parents caring for a short-statured Child – Re- 3 view and Recommendations for Future Research

The study, for the most part, is well-written and clear on its procedures and on the study’s limitations. However, the results of the study are extremely misleading. If a search of Google Scholar is conducted using the same parameters as did the authors, over 500,000 articles are returned on this topic. Looking over the first twenty pages of this search, all of the articles that come up are relevant. In other words, it is untrue that there has been little research on this topic. There are many point the authors make that require further explanation. As well, many of the claims made do not have supporting references. There is an error, either on the bar graph or in the text.

For this paper to be publishable, the authors have to be very clear that their study does not represent the full range of articles that have actually been published on the topic. Major revisions would be required to make this paper publishable.

1.       Why only Using pre-defined inclusion and exclusion criteria, we systematically 12 searched for literature using PubMed and Web of Science. Why only this web site been taken. Any justification

2.       Some article the age of the children are 11 and some are 14.any justification of this type of journal article

3.       What is the main idea for introduction, method, and the findings

4.       What is the main implication of those literature

5.       Very limited article been selected to draw conclusion

Author Response

Reviewer 2:

The study, for the most part, is well-written and clear on its procedures and on the study's limitations. However, the results of the study are extremely misleading. If a search of Google Scholar is conducted using the same parameters as did the authors, over 500,000 articles are returned on this topic. Looking over the first twenty pages of this search, all of the articles that come up are relevant. In other words, it is untrue that there has been little research on this topic. There are many point the authors make that require further explanation. As well, many of the claims made do not have supporting references. There is an error, either on the bar graph or in the text.

For this paper to be publishable, the authors have to be very clear that their study does not represent the full range of articles that have actually been published on the topic. Major revisions would be required to make this paper publishable.

Author's answer: Dear Reviewer, thank you for your detailed feedback. We got the impression that it may not have been entirely clear that we focused our topic on the parental burdens and the parents' HRQOL, and there are certainly not 500,000 hits related to this topic. Your feedback was quite general, making it difficult to address your specific demands in detail. However, we still made an effort to meet your expectations and hope that we could emphasize more clearly that we examined the parental burdens and HRQOL in the studies. As a result, we found all relevant articles on PubMed and Web of Science regarding this topic.

Reviewer comment # 1: Why only Using pre-defined inclusion and exclusion criteria, we systematically 12 searched for literature using PubMed and Web of Science. Why only this web site been taken. Any justification

Author's answer: Dear Reviewer, thank you for suggesting other databases and asking for our reasons for using pre-defined inclusion and exclusion criteria.

We employed pre-defined inclusion and exclusion criteria to promote transparency and objectivity in our review. Furthermore, we aimed to conduct a robust and reliable analysis of all relevant articles pertaining to our topic and ensure reproducibility. We conducted our search only on PubMed and Web of Science because PubMed alone is a highly reputable and comprehensive peer-reviewed database. PubMed offers a detailed search function that effectively meets our search query requirements. Web of Science was an additional source, yielding a few extra results. This further demonstrates that PubMed can effectively cover our research area. In addition, we attempted to expand our results by exploring the references of other articles on the topic. However, it quickly became apparent that we had identified all relevant publications on parental burdens in children with IGHD or ISS.

Reviewer comment # 2: Some article the age of the children are 11 and some are 14.any justification of this type of journal article

Author's answer: Dear Reviewer, thank you for your feedback and the request for clarification. In our pre-defined inclusion and exclusion criteria, we specified that we will include all studies/publications in which the authors present parental quality of life and the burden of parents with children aged 0 to 21. In this way, we wanted to cover the phase of intensive care work with minor children and, at the same time, add the transition period (we defined this as up to 21 years) since children with chronic diseases are often cared for longer by their parents. Therefore, it is logical that the age of the children varies. We had no restrictions regarding the age of the parents.

Reviewer comment # 3: What is the main idea for introduction, method, and the findings

Author's answer: Dear Reviewer, thank you for your feedback and interest in understanding our main idea for introduction, method, and results. The main objective of this study is to demonstrate that parents of children with IGHD/ISS may experience challenges in caring for their children, such as parental burden due to social stigmatization or the administration of injections. We aimed to highlight that these parental burdens and resulting needs are a relevant topic and that a standardized diagnostic tool should urgently be developed to assess these parents' caregiving burdens and needs and thus provide assistance to those in need. While some authors have attempted to address these issues, their approaches differ significantly and often focus on specific aspects, such as parental anxiety in IGHD/ISS children. Therefore, this review reveals the potential challenges in the care of IGHD/ISS children by their parents and the lack of a standardized and disease-specific diagnostic tool for this purpose.

Reviewer comment # 4: What is the main implication of those literature

Author's answer: Dear Reviewer, thank you for your feedback and the request to present the main implications of the literature. The main conclusion is presented in the conclusion section. Moreover, we have highlighted the critical statements once again. The main finding of the included literature is that parents of IGHD/ISS children can experience a variety of burdens in dealing with their child's chronic condition and often have a compromised HRQOL. The parental burdens include difficulties organizing the treatment with growth hormone injections, social pressure, and stigmatization. Concerns about the future also greatly affected parents in most studies, and some studies indicated high stress and anxiety levels among parents. Overall, the included studies demonstrate that there can be a multitude of parental burdens in dealing with IGHD/ISS children, significantly impacting the quality of life of both parents and the entire family.

Reviewer comment # 5: Very limited article been selected to draw conclusion

Author's answer: Dear Reviewer, thank you for your feedback on our conclusion. Our review focused on parental burdens and HRQOL, and the relevant studies were identified and discussed. The conclusion pertains to the implications of the studies, the reported HRQOL of parents, and the assessment of parental HRQOL and burdens. However, we have added additional aspects to the discussion and the conclusion to contextualize the results within the broader framework of parenthood in general and parenting in chronic conditions.  

Reviewer comment # 6: For this paper to be publishable, the authors have to be very clear that their study does not represent the full range of articles that have actually been published on the topic.

Author's answer: Dear Reviewer, thank you for your feedback on our review in total. Our review aims to provide a valuable overview of the current state of studies on the HRQOL and the burdens of parents with children with short stature, using the GHD and ISS diagnoses as examples. We focus exclusively on the perspective of the parents. We do not intend this review to describe children's quality of life or provide an overview of clinical aspects and current therapeutic approaches for the treatment of ISS and GHD. The search for publications was conducted comprehensively, and the conclusion is based solely on the publications identified in the review process.

Reviewer 3 Report

The manuscript presents an interesting theme, however, I believe that the paper should be rewritten to be published.

I present some considerations:

abstract: the abstract should be attractive to the reader. What is presented is very superficial and not very descriptive.

introduction: I believe that the most described theoretical approach could help the reader to understand the context to be researched and also the implications of the research for the community. The introduction has important information, but there is a lack of studies that demonstrate the importance of the topic and also highlight the lack of research.

Method: It should be further described. There should be a description of the study design. There must also be a report in terms of materials used (descriptors, databases, filters used). The inclusion and exclusion criteria should be further described. Procedures must be thoroughly reported to be reproducible. The authors used the prism but this should come as an explanation of how it was done.

There is no data analysis.

Results: The results must be rewritten. There is a quantitative part that could be further explored and an extensive qualitative part, which is not clear and does not clarify the study's objectives for the reader. The data could be grouped and the studies that bring some additional contribution could be cited.

There is no discussion. The conclusion should also be rewritten as it is superficial and does not bring relevant contributions.

Author Response

Reviewer 3:

The manuscript presents an interesting theme, however, I believe that the paper should be rewritten to be published.

Author's answer: Dear Reviewer, thank you for your assessment and feedback on our review. We have attempted to revise the text passages as you suggested, and in the following, we will address your comments.

Reviewer comment # 1: Abstract: The abstract should be attractive to the reader. What is presented is very superficial and not very descriptive.

Author's answer: Dear Reviewer, thank you for evaluating the abstract. We assumed our abstract was kept scientifically simple while presenting the essential methodologies and findings. Nonetheless, we have revised it again to make it more appealing and informative to the reader.

Reviewer comment # 2: Introduction: I believe that the most described theoretical approach could help the reader to understand the context to be researched and also the implications of the research for the community. The introduction has important information, but there is a lack of studies that demonstrate the importance of the topic and also highlight the lack of research.

Author's answer: Dear Reviewer, thank you for assessing the introduction. We have revised the introduction and added additional references to highlight the relevance of the topic and demonstrate the scarcity of research on the parental burden of IGHD/ISS children (see lines 40 to 51): " In 2019 it got estimated that 144 million children worldwide and under five years were short-statured, according to United Nations Children’s Fund (UNICEF), World Health Organization [5]. A study that examined the data of the Pfizer International Growth Study (KIGS ®) from Europe, Asia, and Japan revealed that 46.9% of the sample with short stature (n = 83,803) exhibited short stature as a result of IGHD, while 8.2% presented with ISS [6]. This finding underscores the significance of IGHD and ISS in children within the medical system. Another study highlights that short stature is one of the most frequent concerns pediatric endocrinologists and other physicians caring for children must deal with [4]. Given that this topic affects many children and their families, several studies were conducted to investigate the IGHD/ISS children's well-being and health-related quality of life (HRQOL). This was also reinforced by the fact that the impact of chronic conditions on children and their families has gained importance in recent years [7, 8]."

As well as in lines 78 to 80: "Although numerous studies have been conducted on short stature in general and its impact on children, only a limited number of studies have addressed the impact of pediatric IGHD/ISS on parents, and the results are inconclusive [25]."

Reviewer comment # 3: Method: It should be further described. There should be a description of the study design. There must also be a report in terms of materials used (descriptors, databases, filters used). The inclusion and exclusion criteria should be further described. Procedures must be thoroughly reported to be reproducible. The authors used the prism but this should come as an explanation of how it was done.

Author's answer: Dear Reviewer, thank you for your critical feedback on the method section. The work is a literature search based on the methodological framework of Arksey and O'Malley (2005), which was performed in two databases without any restrictions on publication date or language. The respective section (lines 96-100) now reads: “We conducted a literature review in December 2022 using PubMed and Web of Science (Core Collection) without limitation to the year of publication, the language employed, or accessibility of full-text articles. Furthermore, sources in the included articles were searched for additional material (hand search). We followed the methodological framework of Arksey and O'Malley [31].”

We present the inclusion and exclusion criteria in Figure 1.

To enable reproducibility, we have added the search term used (lines 104-124) and described the composition of this term in more detail. This section now reads: “Publications up to the search date, including information about HRQOL, caregiving burden, or special needs of parents of IGHD/ISS children, were identified. Text word searches and Mesh-Terms, only available on PubMed, were used to avoid missing relevant articles. We used a combination of keywords and database-specific search terms. The search term included a combination of keywords and MeSH terms combined with the Boolsche Operator “AND” and “OR”: For the diagnosis ISS, we used the following term: "Growth Disorders"[Mesh] OR "Growth disorder*"[tw] OR "short stature"[tw] OR "idiopathic short stature"[tiab] OR "Dwarfism"[Mesh] OR "rare condition*"[tiab]; for the diagnosis of IGHD we used this term: "Dwarfism, Pituitary"[Mesh] OR "Dwarfism, Pituitary/psychology"[Mesh] OR "Hypopituitarism"[Mesh] OR "Pituitary insufficiency"[tiab] OR "Insulin-Like Growth Factor II/deficiency"[Mesh] OR "Insulin-Like Growth Factor I/deficiency"[Mesh] OR "Insulin-like growth factor-I Deficiency"[tiab] OR "Human Growth Hormone/deficiency"[Mesh] OR "Growth Hormone/deficiency"[Mesh] OR "Growth Hormone-Releasing Hormone/deficiency"[Mesh] OR "growth hormone deficiency" [tw]. Those terms were connected to the following terms with “AND“: "Parents"[Mesh] OR "Parents/psychology"[Mesh] OR "Caregivers"[Mesh] OR Parent*[tw] OR mother*[tw] OR father*[tw] OR caregiver*[tw] AND "Cost of Illness"[Mesh] OR "costs of illness"[tw] OR "Quality of Life"[Mesh] OR "quality of life"[tw] OR "parental quality of life"[tw] OR "Mental Health"[Mesh] OR "mental health"[tw] OR "Parent-Child Relations"[Mesh] OR "Family Conflict"[Mesh] OR "well being "[tw] OR "well being "[tw] OR "emotional drain"[tw] OR "caregiving stress"[tw] OR "caregiving burden"[tiab] OR "parental burden"[tw] OR "parent reported outcome*"[tw] OR "psychosocial outcome"[tw] OR "psychosocial need*" [tw] OR "burden of disease"[tw] OR "health related quality of life"[tw] OR "health outcome"[tw].”

Furthermore we added information about the selection process to describe the PRISMA-Flow-Chart. The respective section now reads (lines 125 to 142): “The process of publication selection followed the PRISMA statement [32]. We used pre-defined inclusion and exclusion criteria to screen titles and abstracts (LL) (Figure 1). Included in this study were research papers that encompassed parents of children diagnosed with IGHD or ISS within the age range of 0-21 years. Studies examining the QoL, mental health, or general well-being of parents were considered for inclusion, as well as those investigating the parental burdens associated with their child's chronic condition. Additionally, studies exploring parental needs and resources were included. Furthermore, the inclusion criteria involved peer-reviewed journals, as well as cross-sectional studies, clinical trials, prospective studies, longitudinal studies, qualitative studies, and case reports. Excluded from this study were research papers that focused on parents of children with causes of short stature other than IGHD/ISS, such as Achondroplasia, small for gestational age, Turner syndrome, skeletal dysplasia, or psychosocial dwarfism. The only exception was made for studies that included the largest number of participants who were parents of children with IGHD/ISS and evaluated them separately. Studies that solely compared treatment and non-treatment groups, with a focus on the effectiveness of growth hormone therapy, were also excluded. Additionally, studies that solely examined the child's HRQOL were excluded. Conference abstracts, reviews, and meta-analyses were also excluded from the study.”

Reviewer comment # 4: There is no data analysis.

Author's answer: Dear Reviewer, thank you for your feedback. Our goal is to provide an overview of the current state of research on the HRQOL and the burdens of parents with short-statured children, using the diagnoses of ISS and GHD as examples. We did not intend to conduct a meta-analysis. Therefore, we focus on a detailed description of the identified publications: For the quantitative publications, we looked more closely at the patient-reported outcome measures used to determine which tools may be suitable for routine care.

Reviewer comment # 5: Results: The results must be rewritten. There is a quantitative part that could be further explored and an extensive qualitative part, which is not clear and does not clarify the study's objectives for the reader. The data could be grouped and the studies that bring some additional contribution could be cited.

Author's answer: Dear Reviewer, thank you for your assessment. In the results section, we describe the identified publications in detail. We present publication of a study group jointly, as they are based on the same context. In addition, we have structured the studies concerning their main results. So publications that are presented first describe a restricted HRQOL and massive stress for parents of short-statured children. After this, we present the publications in which no or few restrictions are reported. The embedding in the overall context of parental stress only occurs in the discussion section since the results section only presents the review results. A detailed quantitative data analysis was not intended and is therefore not presented. A meta-analysis had a different focus, either related to comparisons with reference populations or examining the impact of interventions.

Reviewer comment # 6: There is no discussion. The conclusion should also be rewritten as it is superficial and does not bring relevant contributions.

Author's answer: Dear Reviewer, thank you for the critical review of the conclusion. We have modified our conclusion to emphasize the parental role and its significance in the development of chronically ill children (see lines 648 to 665).

The respective section now reads (lines 519-541): Parenthood represents an incisive and momentous experience. The transition to parenthood will be assumed not only as a crisis but as well as a normative event; it is often underestimated [49]. At that it has to be noticed that parenthood is a decades-long developmental task, resulting in various challenges. Especially parents/primary caregivers of young children experience a relinquishment of autonomy, personal liberty, occupational identity, and social and leisure activities [50]. In addition to increased requirements [51], mothers reported multiple changes in their lives, which were experienced as stressful [51-53]. These challenges include mental health costs, such as time, physical and emotional well-being, conflicts of social roles, and economic restrictions [51-53]. Additionally, aspects of, e.g., the change of responsibility and feelings of loss [54] were mentioned.

With the diagnosis of a child’s chronic health condition, parents/primary caregivers are confronted with additional responsibilities and tasks that may affect their quality of life [23, 55]. Parents/primary caregivers of a child diagnosed with a chronic health condition may experience higher levels of emotional pressure than parents caring for a healthy child. Next to the primary responsibilities of parents/primary caregivers, they have to deal with additional disease-specific tasks to enable healthy development for their child. Some studies examining the situation of families with a pediatric chronic disease highlighted that these parents have to adjust to restrictions in their social life, job perspective, and psychosocial well-being as their everyday life is decisively determined by the child’s care responsibilities [23, 55]. The parental adaptation to the new situation of having a chronically ill child often happens without much support for coping with the child’s diagnosis. Supplies of psychosocial support are rarely offered, making it hard to assimilate the new family situation [55].”

Reviewer 4 Report

The article titled " Health-related Quality of Life, Stress, Caregiving Burden and Special Needs of Parents caring for a short-statured Child – Review and Recommendations for Future Research" is of great interest. However, I would like to make some comments in order to improve the article.

-Review the citations in the text and adapt them to the rules of the journal

-¿Why do you only review PubMed and Web of Science?

-I would recommend introducing some figure in the results section that summarizes them or Ii would be interesing to differentiate in sections the different aspects addressed

-The discussion repeats some of the issues raised in the results

-The conclusions are very short and do not include the limitations of the study

Author Response

Reviewer 1:

Reviewer comment # 1: The manuscript gives clear picture in understanding the caregiving burden of the parents of children with short stature. The following comments are made to improve the quality of the study for publication.

Author's answer: Dear Reviewer, thank you for your feedback on our review. Following your suggestions, we have optimized the manuscript in the following manner.

Reviewer comment # 2: Introduction: Correct the sentence on line 27 and 28 to give clarity on linking Isolated growth hormone deficiency (IGHD) to short stature.

Author's answer: Dear Reviewer, thank you for your critical review. We have optimized the sentence to give more clarity. The respective section now reads (lines 33 to 37): " It is associated with many different diseases, like genetic or endocrine ones [1], whereas the most common causes are growth hormone deficiency, hypothyroidism, Turner syndrome, and celiac disease [2]. Isolated growth hormone deficiency (IGHD) is an endocrine disease caused by a lack or insufficiency of growth hormone (GH) secretion [3] and therefore leads to short stature. "

Reviewer comment # 3: Introduction: Correct the statement in line 36 to 40 to give better understanding.

Author's answer: Dear Reviewer, thank you for your feedback. We have corrected the sentence. It now reads (lines 48 to 56): "Given that this topic affects many children and their families, several studies were conducted to investigate the IGHD/ISS children's well-being and health-related quality of life (HRQOL). This was also reinforced by the fact that the impact of chronic conditions on children and their families has gained importance in recent years [7, 8]. Most studies assume that short-statured children have a lower HRQOL compared to their normal-statured peers [9] and have low self-esteem [10], primarily because of short-statured children being bullied at school. Worries about their height, feeling inferior about their shortness, and negative comparison with peers were additional results retrieved in the studies on IGHD/ISS children's HRQOL [11-13]. "

Reviewer comment # 4: Introduction: Rearrange the statement on line 46-47 to give better understanding.

Author's answer: Dear Reviewer, thank you for your hint on this section. We have rearranged the statement (lines 58 to 68): " Especially the parents of children with a chronic condition are often responsible for maintaining the functioning of the family by emphasizing the positive aspects of development of their children and helping them cope with their chronic illness [16, 17]. Socioecological factors like a good functioning parent-child relationship and parental adaptation are some of the main protective factors for these children [18, 19]. Moreover, studies have shown that high caregiving stress levels can affect children by causing depression, anxiety, and feeling desolate [20]. Therefore, it is crucial to investigate parental HRQOL, burdens, and special needs throughout their children's short stature and to understand how they are affected by caring for a short-statured child because parents can modulate their children's intrapersonal emotional attitude towards themselves with social support and coping strategies [21, 22]."

Reviewer comment # 5: Materials and Methods: Elaborate on statement 74 in relation to the statement in line 71 regarding without limitations.

Author's answer: Dear Reviewer, thank you for your critical review. We have corrected the statement as follows (see lines 96 to 100): "We conducted a literature review in December 2022 using PubMed and Web of Science (Core Collection) without limitation to the year of publication, the language employed, or accessibility of full-text articles. Furthermore, sources in the included articles were searched for additional material (hand search). We followed the methodological framework of Arksey and O'Malley [31]."

Reviewer comment # 6: Materials and Methods: Provide clarity in line 76 and consider removing the brackets. Elaborate without using the brackets.

Author's answer: Dear Reviewer, thank you for the hint on this line. We have removed the brackets (lines 102 to 104): " Text word searches and Mesh-Terms, only available on PubMed, were used to avoid missing relevant articles. We used a combination of keywords and database-specific search terms. "

Reviewer comment # 7: Materials and Methods: Clarify LL and SW bracketed words in lines 79 and 80.

Author's answer: Dear Reviewer, thank you for the indication. We have clarified these lines (see lines 145 to 146): "Two independent raters (Lea Lackner and Stefanie Witt) screened eligible full texts to ensure an unbiased selection."

Reviewer comment # 8: Results: The detail report of each study included is presented.

Author's answer: Dear Reviewer, thank you for this feedback. We intended to present the results precisely and clearly.

Reviewer comment # 9: Funding – please clarify the statement in line 534

Author's answer: Dear Reviewer, thank you for this hint. We have clarified the statement (see line 645): "This research received no external funding.”

Reviewer 2:

The study, for the most part, is well-written and clear on its procedures and on the study's limitations. However, the results of the study are extremely misleading. If a search of Google Scholar is conducted using the same parameters as did the authors, over 500,000 articles are returned on this topic. Looking over the first twenty pages of this search, all of the articles that come up are relevant. In other words, it is untrue that there has been little research on this topic. There are many point the authors make that require further explanation. As well, many of the claims made do not have supporting references. There is an error, either on the bar graph or in the text.

For this paper to be publishable, the authors have to be very clear that their study does not represent the full range of articles that have actually been published on the topic. Major revisions would be required to make this paper publishable.

Author's answer: Dear Reviewer, thank you for your detailed feedback. We got the impression that it may not have been entirely clear that we focused our topic on the parental burdens and the parents' HRQOL, and there are certainly not 500,000 hits related to this topic. Your feedback was quite general, making it difficult to address your specific demands in detail. However, we still made an effort to meet your expectations and hope that we could emphasize more clearly that we examined the parental burdens and HRQOL in the studies. As a result, we found all relevant articles on PubMed and Web of Science regarding this topic.

Reviewer comment # 1: Why only Using pre-defined inclusion and exclusion criteria, we systematically 12 searched for literature using PubMed and Web of Science. Why only this web site been taken. Any justification

Author's answer: Dear Reviewer, thank you for suggesting other databases and asking for our reasons for using pre-defined inclusion and exclusion criteria.

We employed pre-defined inclusion and exclusion criteria to promote transparency and objectivity in our review. Furthermore, we aimed to conduct a robust and reliable analysis of all relevant articles pertaining to our topic and ensure reproducibility. We conducted our search only on PubMed and Web of Science because PubMed alone is a highly reputable and comprehensive peer-reviewed database. PubMed offers a detailed search function that effectively meets our search query requirements. Web of Science was an additional source, yielding a few extra results. This further demonstrates that PubMed can effectively cover our research area. In addition, we attempted to expand our results by exploring the references of other articles on the topic. However, it quickly became apparent that we had identified all relevant publications on parental burdens in children with IGHD or ISS.

Reviewer comment # 2: Some article the age of the children are 11 and some are 14.any justification of this type of journal article

Author's answer: Dear Reviewer, thank you for your feedback and the request for clarification. In our pre-defined inclusion and exclusion criteria, we specified that we will include all studies/publications in which the authors present parental quality of life and the burden of parents with children aged 0 to 21. In this way, we wanted to cover the phase of intensive care work with minor children and, at the same time, add the transition period (we defined this as up to 21 years) since children with chronic diseases are often cared for longer by their parents. Therefore, it is logical that the age of the children varies. We had no restrictions regarding the age of the parents.

Reviewer comment # 3: What is the main idea for introduction, method, and the findings

Author's answer: Dear Reviewer, thank you for your feedback and interest in understanding our main idea for introduction, method, and results. The main objective of this study is to demonstrate that parents of children with IGHD/ISS may experience challenges in caring for their children, such as parental burden due to social stigmatization or the administration of injections. We aimed to highlight that these parental burdens and resulting needs are a relevant topic and that a standardized diagnostic tool should urgently be developed to assess these parents' caregiving burdens and needs and thus provide assistance to those in need. While some authors have attempted to address these issues, their approaches differ significantly and often focus on specific aspects, such as parental anxiety in IGHD/ISS children. Therefore, this review reveals the potential challenges in the care of IGHD/ISS children by their parents and the lack of a standardized and disease-specific diagnostic tool for this purpose.

Reviewer comment # 4: What is the main implication of those literature

Author's answer: Dear Reviewer, thank you for your feedback and the request to present the main implications of the literature. The main conclusion is presented in the conclusion section. Moreover, we have highlighted the critical statements once again. The main finding of the included literature is that parents of IGHD/ISS children can experience a variety of burdens in dealing with their child's chronic condition and often have a compromised HRQOL. The parental burdens include difficulties organizing the treatment with growth hormone injections, social pressure, and stigmatization. Concerns about the future also greatly affected parents in most studies, and some studies indicated high stress and anxiety levels among parents. Overall, the included studies demonstrate that there can be a multitude of parental burdens in dealing with IGHD/ISS children, significantly impacting the quality of life of both parents and the entire family.

Reviewer comment # 5: Very limited article been selected to draw conclusion

Author's answer: Dear Reviewer, thank you for your feedback on our conclusion. Our review focused on parental burdens and HRQOL, and the relevant studies were identified and discussed. The conclusion pertains to the implications of the studies, the reported HRQOL of parents, and the assessment of parental HRQOL and burdens. However, we have added additional aspects to the discussion and the conclusion to contextualize the results within the broader framework of parenthood in general and parenting in chronic conditions.  

Reviewer comment # 6: For this paper to be publishable, the authors have to be very clear that their study does not represent the full range of articles that have actually been published on the topic.

Author's answer: Dear Reviewer, thank you for your feedback on our review in total. Our review aims to provide a valuable overview of the current state of studies on the HRQOL and the burdens of parents with children with short stature, using the GHD and ISS diagnoses as examples. We focus exclusively on the perspective of the parents. We do not intend this review to describe children's quality of life or provide an overview of clinical aspects and current therapeutic approaches for the treatment of ISS and GHD. The search for publications was conducted comprehensively, and the conclusion is based solely on the publications identified in the review process.

Reviewer 3:

The manuscript presents an interesting theme, however, I believe that the paper should be rewritten to be published.

Author's answer: Dear Reviewer, thank you for your assessment and feedback on our review. We have attempted to revise the text passages as you suggested, and in the following, we will address your comments.

Reviewer comment # 1: Abstract: The abstract should be attractive to the reader. What is presented is very superficial and not very descriptive.

Author's answer: Dear Reviewer, thank you for evaluating the abstract. We assumed our abstract was kept scientifically simple while presenting the essential methodologies and findings. Nonetheless, we have revised it again to make it more appealing and informative to the reader.

Reviewer comment # 2: Introduction: I believe that the most described theoretical approach could help the reader to understand the context to be researched and also the implications of the research for the community. The introduction has important information, but there is a lack of studies that demonstrate the importance of the topic and also highlight the lack of research.

Author's answer: Dear Reviewer, thank you for assessing the introduction. We have revised the introduction and added additional references to highlight the relevance of the topic and demonstrate the scarcity of research on the parental burden of IGHD/ISS children (see lines 40 to 51): " In 2019 it got estimated that 144 million children worldwide and under five years were short-statured, according to United Nations Children’s Fund (UNICEF), World Health Organization [5]. A study that examined the data of the Pfizer International Growth Study (KIGS ®) from Europe, Asia, and Japan revealed that 46.9% of the sample with short stature (n = 83,803) exhibited short stature as a result of IGHD, while 8.2% presented with ISS [6]. This finding underscores the significance of IGHD and ISS in children within the medical system. Another study highlights that short stature is one of the most frequent concerns pediatric endocrinologists and other physicians caring for children must deal with [4]. Given that this topic affects many children and their families, several studies were conducted to investigate the IGHD/ISS children's well-being and health-related quality of life (HRQOL). This was also reinforced by the fact that the impact of chronic conditions on children and their families has gained importance in recent years [7, 8]."

As well as in lines 78 to 80: "Although numerous studies have been conducted on short stature in general and its impact on children, only a limited number of studies have addressed the impact of pediatric IGHD/ISS on parents, and the results are inconclusive [25]."

Reviewer comment # 3: Method: It should be further described. There should be a description of the study design. There must also be a report in terms of materials used (descriptors, databases, filters used). The inclusion and exclusion criteria should be further described. Procedures must be thoroughly reported to be reproducible. The authors used the prism but this should come as an explanation of how it was done.

Author's answer: Dear Reviewer, thank you for your critical feedback on the method section. The work is a literature search based on the methodological framework of Arksey and O'Malley (2005), which was performed in two databases without any restrictions on publication date or language. The respective section (lines 96-100) now reads: “We conducted a literature review in December 2022 using PubMed and Web of Science (Core Collection) without limitation to the year of publication, the language employed, or accessibility of full-text articles. Furthermore, sources in the included articles were searched for additional material (hand search). We followed the methodological framework of Arksey and O'Malley [31].”

We present the inclusion and exclusion criteria in Figure 1.

To enable reproducibility, we have added the search term used (lines 104-124) and described the composition of this term in more detail. This section now reads: “Publications up to the search date, including information about HRQOL, caregiving burden, or special needs of parents of IGHD/ISS children, were identified. Text word searches and Mesh-Terms, only available on PubMed, were used to avoid missing relevant articles. We used a combination of keywords and database-specific search terms. The search term included a combination of keywords and MeSH terms combined with the Boolsche Operator “AND” and “OR”: For the diagnosis ISS, we used the following term: "Growth Disorders"[Mesh] OR "Growth disorder*"[tw] OR "short stature"[tw] OR "idiopathic short stature"[tiab] OR "Dwarfism"[Mesh] OR "rare condition*"[tiab]; for the diagnosis of IGHD we used this term: "Dwarfism, Pituitary"[Mesh] OR "Dwarfism, Pituitary/psychology"[Mesh] OR "Hypopituitarism"[Mesh] OR "Pituitary insufficiency"[tiab] OR "Insulin-Like Growth Factor II/deficiency"[Mesh] OR "Insulin-Like Growth Factor I/deficiency"[Mesh] OR "Insulin-like growth factor-I Deficiency"[tiab] OR "Human Growth Hormone/deficiency"[Mesh] OR "Growth Hormone/deficiency"[Mesh] OR "Growth Hormone-Releasing Hormone/deficiency"[Mesh] OR "growth hormone deficiency" [tw]. Those terms were connected to the following terms with “AND“: "Parents"[Mesh] OR "Parents/psychology"[Mesh] OR "Caregivers"[Mesh] OR Parent*[tw] OR mother*[tw] OR father*[tw] OR caregiver*[tw] AND "Cost of Illness"[Mesh] OR "costs of illness"[tw] OR "Quality of Life"[Mesh] OR "quality of life"[tw] OR "parental quality of life"[tw] OR "Mental Health"[Mesh] OR "mental health"[tw] OR "Parent-Child Relations"[Mesh] OR "Family Conflict"[Mesh] OR "well being "[tw] OR "well being "[tw] OR "emotional drain"[tw] OR "caregiving stress"[tw] OR "caregiving burden"[tiab] OR "parental burden"[tw] OR "parent reported outcome*"[tw] OR "psychosocial outcome"[tw] OR "psychosocial need*" [tw] OR "burden of disease"[tw] OR "health related quality of life"[tw] OR "health outcome"[tw].”

Furthermore we added information about the selection process to describe the PRISMA-Flow-Chart. The respective section now reads (lines 125 to 142): “The process of publication selection followed the PRISMA statement [32]. We used pre-defined inclusion and exclusion criteria to screen titles and abstracts (LL) (Figure 1). Included in this study were research papers that encompassed parents of children diagnosed with IGHD or ISS within the age range of 0-21 years. Studies examining the QoL, mental health, or general well-being of parents were considered for inclusion, as well as those investigating the parental burdens associated with their child's chronic condition. Additionally, studies exploring parental needs and resources were included. Furthermore, the inclusion criteria involved peer-reviewed journals, as well as cross-sectional studies, clinical trials, prospective studies, longitudinal studies, qualitative studies, and case reports. Excluded from this study were research papers that focused on parents of children with causes of short stature other than IGHD/ISS, such as Achondroplasia, small for gestational age, Turner syndrome, skeletal dysplasia, or psychosocial dwarfism. The only exception was made for studies that included the largest number of participants who were parents of children with IGHD/ISS and evaluated them separately. Studies that solely compared treatment and non-treatment groups, with a focus on the effectiveness of growth hormone therapy, were also excluded. Additionally, studies that solely examined the child's HRQOL were excluded. Conference abstracts, reviews, and meta-analyses were also excluded from the study.”

Reviewer comment # 4: There is no data analysis.

Author's answer: Dear Reviewer, thank you for your feedback. Our goal is to provide an overview of the current state of research on the HRQOL and the burdens of parents with short-statured children, using the diagnoses of ISS and GHD as examples. We did not intend to conduct a meta-analysis. Therefore, we focus on a detailed description of the identified publications: For the quantitative publications, we looked more closely at the patient-reported outcome measures used to determine which tools may be suitable for routine care.

Reviewer comment # 5: Results: The results must be rewritten. There is a quantitative part that could be further explored and an extensive qualitative part, which is not clear and does not clarify the study's objectives for the reader. The data could be grouped and the studies that bring some additional contribution could be cited.

Author's answer: Dear Reviewer, thank you for your assessment. In the results section, we describe the identified publications in detail. We present publication of a study group jointly, as they are based on the same context. In addition, we have structured the studies concerning their main results. So publications that are presented first describe a restricted HRQOL and massive stress for parents of short-statured children. After this, we present the publications in which no or few restrictions are reported. The embedding in the overall context of parental stress only occurs in the discussion section since the results section only presents the review results. A detailed quantitative data analysis was not intended and is therefore not presented. A meta-analysis had a different focus, either related to comparisons with reference populations or examining the impact of interventions.

Reviewer comment # 6: There is no discussion. The conclusion should also be rewritten as it is superficial and does not bring relevant contributions.

Author's answer: Dear Reviewer, thank you for the critical review of the conclusion. We have modified our conclusion to emphasize the parental role and its significance in the development of chronically ill children (see lines 648 to 665).

The respective section now reads (lines 519-541): Parenthood represents an incisive and momentous experience. The transition to parenthood will be assumed not only as a crisis but as well as a normative event; it is often underestimated [49]. At that it has to be noticed that parenthood is a decades-long developmental task, resulting in various challenges. Especially parents/primary caregivers of young children experience a relinquishment of autonomy, personal liberty, occupational identity, and social and leisure activities [50]. In addition to increased requirements [51], mothers reported multiple changes in their lives, which were experienced as stressful [51-53]. These challenges include mental health costs, such as time, physical and emotional well-being, conflicts of social roles, and economic restrictions [51-53]. Additionally, aspects of, e.g., the change of responsibility and feelings of loss [54] were mentioned.

With the diagnosis of a child’s chronic health condition, parents/primary caregivers are confronted with additional responsibilities and tasks that may affect their quality of life [23, 55]. Parents/primary caregivers of a child diagnosed with a chronic health condition may experience higher levels of emotional pressure than parents caring for a healthy child. Next to the primary responsibilities of parents/primary caregivers, they have to deal with additional disease-specific tasks to enable healthy development for their child. Some studies examining the situation of families with a pediatric chronic disease highlighted that these parents have to adjust to restrictions in their social life, job perspective, and psychosocial well-being as their everyday life is decisively determined by the child’s care responsibilities [23, 55]. The parental adaptation to the new situation of having a chronically ill child often happens without much support for coping with the child’s diagnosis. Supplies of psychosocial support are rarely offered, making it hard to assimilate the new family situation [55].”

Reviewer 4:

Reviewer comment # 1: Review the citations in the text and adapt them to the rules of the journal

Author's answer: Dear Reviewer, thank you for your hint on the citation style. We have checked our citations and the reference list. We now used the EndNote Style “MDPI” provided as a download by the target journal to ensure the correct citation style.

Reviewer comment # 2: Why do you only review PubMed and Web of Science?

Author's answer: Dear Reviewer, thank you for your feedback. We conducted our search only on PubMed and Web of Science because PubMed alone is a highly reputable and comprehensive peer-reviewed database. PubMed offers a detailed search function that effectively meets our search query requirements. Web of Science was an additional source, yielding a few extra results. This further demonstrates that PubMed can effectively cover our research area. In addition, we attempted to expand our results by exploring the references of other articles on the topic. However, it quickly became apparent that we had identified all relevant articles on parental burdens in children with IGHD or ISS.

Reviewer comment # 3: I would recommend introducing some figure in the results section that summarizes them or Ii would be interesing to differentiate in sections the different aspects addressed

Author's answer: Dear Reviewer, thank you for suggesting rearranging the included publications. We have restructured the presentaion of the included publications. We present the publications of a study group jointly, as they are based on the same context. In addition, we have structured the studies concerning their main results. So publications that are presented first describe a restricted HRQOL and massive stress for parents of short-statured children. After this, we present the publications in which no or few restrictions are reported.

Reviewer comment # 4: The discussion repeats some of the issues raised in the results

Author's answer: Dear Reviewer, thank you for critically reviewing our discussion. In the discussion, we have again summarized the key findings of the studies to highlight them and facilitate comparison. We aimed to simplify the content for the reader to follow the argumentation and grasp the main points at a glance. Nevertheless, we have added several points to our discussion section, such as the broader framework of parenthood in general and parenting in the context of chronic conditions.  

Reviewer comment # 5: The conclusions are very short and do not include the limitations of the study

Author's answer: Dear Reviewer, thank you for this hint. We added more content to the conclusion section. The respective section now reads (lines 637 to 654): “Parenthood itself is a decades-long task with various challenges. Parents/primary caregivers of children with chronic health conditions are confronted with additional challenges. While IGHD and ISS are not life-threatening health conditions, these challenges are often understated, and parents/primary caregivers rarely seek help in processing the diagnosis and in day-to-day tasks. In contrast, this review points out that parents of IGHD/ISS children can have various burdens, anxiety, special needs, and an impaired HRQOL in caring for their short-statured child. Clinicians should keep in mind that parental support may positively affect the child’s development.

On top of that, our results support earlier findings of the limited use of disease-specific measures to assess these domains. Still, most measurements are generic, hence not very sensitive. However, the Brod Group has contributed towards disease-specific measures for parents of IGHD/ISS children with the development of the GHD-PTB. Further studies should aim to develop measures for all domains of affected parents’ lives. For children, especially those with a chronic condition, family support is a main protective factor and helps children cope with their disease. Therefore, assessing the caregiving burden, HRQOL, and special needs of parents and families of IGHD/ISS children is crucial. Because once the parents/families are adapted to the child’s short stature, they can increase their child’s health-related outcomes and well-being.”

Furthermore, we have also added a section on limitations at the end of the discussion section. The new section reads (lines 622 to 635): " Limitations of this study include that we only used two databases for our search. While we believe that we identified all relevant articles through our search, we cannot definitively exclude the possibility that including additional databases would have yielded further results. Additionally, we did not assess the quality of the studies, making it challenging to compare the findings. Moreover, the included studies had very different study designs that were not directly comparable. Furthermore, the included studies primarily focused on mothers, making it difficult to generalize the results to parents in general. Comparing the results is further complicated because the studies were conducted in different healthcare systems. Cultural differences are also likely to influence the assessment of burdens and concerns. Most publications were from Europe and the USA, with only one from Sri Lanka. Another limitation of our review is that the included studies employed different recruitment strategies, often purposive sampling, which introduces a potential bias. Nonetheless, our review highlights parental burdens in dealing with IGHD/ISS children and draws attention to the limited research available on this topic.”

Round 2

Reviewer 3 Report

First, I would like to congratulate the authors for their efforts to improve the manuscript.

I present, below, some suggestions for the manuscript:

Method: clarify the research design;

clarify what would be the methodology of Arksey and O'Malley [31].

remove the names of researchers who helped from the body of the text (Lea Lackner and Stefanie Wit- line 145)

add data analysis

Results: the authors could compile the results and highlight what would be most important in each study, instead of describing each one - the reading becomes very dense and hard to understand.

Author Response

Reviewer 3:

First, I would like to congratulate the authors for their efforts to improve the manuscript.

Author's answer: Thank you very much. We have tried to implement the suggested changes and requests as best as possible and think we improved the manuscript significantly.

I present, below, some suggestions for the manuscript:

Author's answer: We appreciate the renewed critical review of the revised manuscript and thank you for the constructive feedback. We have highlighted our edits in color and shown each aspect in the point-to-point response.

Reviewer comment # 1: Method: clarify the research design; clarify what would be the methodology of Arksey and O'Malley [31].

Author's answer: Dear Reviewer, thank you for this hint. Based on your suggestions, we specified the aim of our study, namely conducting a scoping review to understand the aspects of caregiving burden in short-stature youth and identifying patient-reported outcomes measures assessing caregiving burden and parental quality of life.

The respective section now reads (lines 89-92): “Therefore, we aim to understand the aspects of the caregiving burden in short-stature youth and identify patient-reported outcomes measures (PROMs) assessing caregiving burden and parental quality of life for use in routine care and research by conducting a scoping review.”

Furthermore, we added information about the methodological framework (lines 97-101). “We followed the methodological framework of Arksey and O'Malley [31] for scoping reviews. This framework includes five stages as well as an optional sixth stage: (1) identifying the research question, (2) identifying relevant studies, (3) study selection, (4) charting the data, (5) collating, summarizing, and reporting the results, and (6) a consultation exercise [31].”

Lines 150-152: “We charted the data using Microsoft Excel and identified the relevant data on the sample (population and country of origin), used study design, the aim of the studies and used PROMs, and main results from all included publications.”

Reviewer comment # 2: Remove the names of researchers who helped from the body of the text (Lea Lackner and Stefanie Wit- line 145)

Author's answer: We removed the names from the body of the text (line 146).

Reviewer comment # 3: Add data analysis

Author's answer: Dear reviewer. We understand that analyzing the quantitative data, for example, a meta-analysis would be a quality improvement. Nonetheless, our review aims to describe the burdens and quality of life of parents of short-statured children and adolescents and identify currently used validated PROMs that (can) be used in the clinical context but also research. An analysis of the data beyond this is not the intended goal and will not be conducted here.

Reviewer comment # 4: Results: the authors could compile the results and highlight what would be most important in each study, instead of describing each one - the reading becomes very dense and hard to understand.

Author's answer:

Dear reviewer, thank you for pointing out this weakness in our manuscript. We restructured the results section (lines 178-587). First, we have condensed the results. These now focus exclusively on the burdens on parents. Second, we have added a section on the validated PROMs used in the presented studies. These changes allows the reader to comprehend the discussion and conclusion better.
